

# Assessing knowledge anxiety in researchers: a comprehensive measurement scale

Yu Zhenlei[1,2,*], Lin Song[3], Dong Minyi[4] and He Qiang[3,*]

[1] Shandong University, Jinan, China
[2] Qilu University of Technology (Shandong Academy of Sciences), Jinan, China
[3] Tianjin College of Traditional Chinese Medicine, Tianjing, China
[4] Linyi University, Linyi, China
[*] These authors contributed equally to this work.

## ABSTRACT

**Background**. The rapid pace of knowledge production has introduced a phenomenon termed "knowledge anxiety", a psychological state where researchers feel inadequate in keeping up with emerging information. This state can negatively affect productivity and mental well-being, yet there is no comprehensive tool to measure knowledge anxiety across different research domains.

**Methods**. We employed a mixed-methods approach to develop a multidimensional scale for assessing knowledge anxiety. Initial items were generated through a literature review and qualitative interviews with 313 researchers. After pilot testing, the main study involved 26 participants. The scale was refined through exploratory factor analysis (EFA) and confirmatory factor analysis (CFA) to ensure its structural validity and reliability.

**Results**. EFA resulted in a 16-item scale with four distinct factors: cognitive, emotional, behavioral, and capability-related anxieties. CFA confirmed a strong model fit, with standardized factor loadings between 0.549 and 0.887. The scale demonstrated high reliability, with a composite Cronbach's alpha of 0.883.

**Conclusions**. This newly developed scale offers a reliable and valid measure of knowledge anxiety, providing researchers with a valuable tool to assess the psychological impacts of knowledge overload.

Corresponding author
He Qiang, 1770251916@qq.com,
57125742@qq.comqq.com

# INTRODUCTION

In this age of unparalleled digital information expansion, while access to knowledge has significantly broadened, a paradoxical and pervasive sense of knowledge inadequacy has intensified among researchers, particularly within the realm of environmental science (*Wang, 2018*). This investigation delves into the phenomenon of 'knowledge anxiety', a psychological state marked by apprehension regarding the acquisition, internalization, and generation of knowledge amidst the abundance of information available (*Lu, Ma & Kong, 2020*). Previous studies have predominantly explored knowledge anxiety from singular disciplinary perspectives, offering limited insights into its complex nature (*Liu & Sun,*

*2010*). Our research adopts an interdisciplinary approach, amalgamating psychological theories of anxiety with insights from information science and sociology, to construct a multidimensional scale of knowledge anxiety (*Wolman, 2001*; *Li, Cui & Zhou, 2022*). This scale not only advances our understanding of the psychological impacts of information overload on researchers but also sets the stage for crafting targeted interventions. The methodology, grounded in robust empirical analysis, underscores the importance of addressing knowledge anxiety as a critical factor influencing researchers' mental health and productivity, particularly those dedicated to confronting the pressing environmental challenges of our time (*Cao et al., 2010*).

As of December 2023, a comprehensive search within the Web of Science database for "knowledge anxiety" yielded zero results in the context of environmental science, underscoring an alarming research void (*Li & Cao, 2011*). The absence of research on how knowledge anxiety affects these dedicated professionals hampers our ability to support their mental well-being and, in turn, their capacity to generate innovative solutions for environmental sustainability (*Kuang, 2019*). Recognizing this gap, our study endeavors to illuminate the multifaceted nature of knowledge anxiety among environmental researchers, offering a pioneering analysis that bridges this critical research lacuna (*Zheng & Ying, 2021*). By focusing on the unique pressures faced by environmental scientists in their quest to keep pace with rapidly evolving knowledge landscapes, this research provides valuable insights into the psychological barriers that may impede scientific progress in crucial environmental domains (*Ding, 2019*; *He, Gong & Yan, 2020*).

## MATERIALS & METHODS

### Development of initial items

Drawing from the insights gathered from the study outlined in the document, synthesizing the findings on knowledge anxiety, research data anxiety, doctoral students' knowledge anxiety, and the conceptual framework formed through grounded theory's three-level coding (*Shen & Cai, 2022*; *Sun, He & Hu, 2021*), this study ventures into unraveling the structural dimensions of knowledge anxiety among researchers. It uncovers that knowledge anxiety in researchers fundamentally manifests as an anxious state encountered during the acquisition, internalization, and output phases of knowledge. Researchers grappling with knowledge anxiety exhibit significant disparities in emotional, cognitive, and behavioral aspects compared to their non-anxious counterparts, aligning with the investigations of *Hui, Xiaohu & Xueyan (2022)* into situational anxiety. Their research validates that situational anxiety encompasses emotional arousal, cognitive interference, and somatic behavior, hence positioning researchers' knowledge anxiety as a manifestation of situational anxiety. Leveraging this theoretical framework, the study deconstructs the structural dimensions of researchers' knowledge anxiety. Utilizing open data and interview insights on researchers' knowledge anxiety, it delves deeper into its specific dimensions. Ultimately, it delineates the structural dimensions of researchers' knowledge anxiety as cognitive anxiety, emotional anxiety, and behavioral anxiety, outlining the deconstruction process and dimensions of researchers' knowledge anxiety manifestations (*Zhenlei et al., 2024*). Consequently, a

three-dimensional structural model of researchers' knowledge anxiety is constructed, and the study defines cognitive anxiety, emotional anxiety, and behavioral anxiety as follows:

Cognitive anxiety signifies the perplexity and apprehension researchers face concerning the absorption, internalization, and transformation of academic knowledge. This includes difficulties in comprehension, knowledge scarcity, and cognitive barriers. For instance, challenges in understanding complex concepts in specialized fields, leading to anxiety and unease, as some respondents articulated a daily sense of anxiety and inability to grasp various formulas in literature. Researchers' struggle with knowledge comprehension stems from a lack of foundational knowledge, as echoed by some, highlighting the inadequate grasp of basic knowledge on their research topics and the resultant comprehension difficulties. Moreover, the demand for creative thinking and problem-solving in research work, coupled with difficulties in understanding and knowledge scarcity, obstructs researchers' absorption and innovation of academic knowledge, thereby triggering emotional anxiety such as feelings of loss and doubt.

Emotional anxiety pertains to the autonomous nervous arousal and unpleasant sensations induced by activities related to academic knowledge acquisition, internalization, and output, manifesting as restlessness, doubt, fear, and despair. For example, the overwhelming distress researchers face under the pressure of producing knowledge, as depicted by some researchers' narratives of the agonizing process of writing papers. Additionally, the sense of helplessness and confusion in the face of constraints in research projects further exacerbates researchers' emotional anxiety, underscoring the imperative to mitigate knowledge anxiety among researchers.

Behavioral anxiety encompasses the array of maladaptive behaviors or reactions displayed by researchers when confronted with knowledge anxiety situations. Typically, this is evident in sleep disturbances, avoidance behaviors, knowledge hoarding, and self-consolation attempts to alleviate the feelings of anxiety or avoid uncomfortable emotional experiences. Furthermore, researchers engage in self-consoling behaviors to ease the pressure of knowledge anxiety, such as the cyclic routine of transporting books to and from the library without engaging with the content, merely creating an illusion of diligent study. Yet, such self-consolation only offers transient relief and fails to address the root causes of knowledge anxiety.

The necessary prerequisite for the successful development of the scale is to construct a project pool covering a rich set of items. The initial formulation of items in this chapter is quite diverse. First, it is based on the three dimensions of knowledge anxiety among researchers constructed using grounded theory in the previous section. Relevant statements that can maximally reflect the dimensions' connotations are selected from the original texts. Under the condition of ensuring that the semantics remain unchanged, the selected statements are standardized, processed, and organized in terms of expression. Second, based on the SOR theoretical model extracting some core elements, further refining one measurement dimension of the initial items related to researchers' knowledge anxiety. Third, by referring to the expressions of items in existing scales for research data anxiety and information anxiety (*Wei, Xia & Li, 2018*; *Han et al., 2019*; *Shen et al., 2021*), and

combining them with the content and characteristics of researchers' knowledge anxiety, contextual adaptations are made.

Furthermore, following the principles of scale development and ensuring that each dimension has no fewer than three measurement items (*Wu, 2010*), and referring to the study by *Fabrigar et al. (1999)* it is suggested that each common factor should include at least four measurement variables, possibly up to six, depending on the expected number and characteristics of factors anticipated by researchers when conducting exploratory factor analysis. This approach enhances the accuracy and interpretability of the factor analysis results. Therefore, in the process of scale development, it is essential to ensure that each dimension has no fewer than three measurement items. Through these steps, a preliminary project pool is formed, including four dimensions and 70 initial measurement items for researchers' knowledge anxiety.

## Item consolidation and refinement

To enhance the simplicity and effectiveness of the scale, it is necessary to consolidate and refine the initial measurement items in the project pool. Firstly, six master's students from different disciplines were invited to form two review groups. The review standards included whether the measurement items conformed to conceptual dimensions, whether there were semantic repetitions, and whether the items expressed clarity. Secondly, each group conducted logical, semantic, and language analyses of the initial items. Items that involved multiple dimensions or concepts were split to ensure that each item reflected only one dimension. Items expressing similar or identical meanings were merged to eliminate duplicate or redundant content. Items with unclear or non-standard expressions were modified to better align with the test-taker's expression style and language norms. After completing the review work, the results of the two groups were compared, and items with differing or controversial treatment were excluded. Following this process, 31 initial measurement items were retained in the project pool. Subsequently, to ensure the reliability and scientific validity of the measurement items, five experts with backgrounds and experience in management, cognitive psychology, behavioral psychology, and sensation and perception psychology were invited to evaluate and provide suggestions for the initial items. Based on the experts' opinions, adjustments or deletions were made to some items. Through these steps, a final research personnel's knowledge anxiety initial measurement scale comprising four dimensions and 22 items was formed. The initial scale measurement items are provided in Appendix S1.

## Pilot testing

Upon completing the initial scale development, it is imperative to examine the feasibility and effectiveness of the scale. This section utilizes pilot testing to scrutinize the initial measurement scale. Pilot testing refers to a small-scale trial of the scale before formal data collection, aiming to identify and address potential issues within the scale, such as question comprehension difficulty and response time (*Meng, Chang & Ye, 2016*). Pilot testing enhances the quality and reliability of the scale, providing references for subsequent large-scale formal testing. The general steps of pilot testing include data collection, item analysis, exploratory factor analysis, and reliability analysis.

## Data collection

In the pilot testing phase, this study employed online survey methods to collect data. The research personnel's knowledge anxiety survey questionnaire was uploaded to the Wenjuanxing platform, generating a questionnaire link. The questionnaire was distributed nationwide to research personnel across various disciplines through social media platforms such as WeChat and QQ. The survey questionnaire comprised the basic information of respondents and the 22 retained measurement items assessed by experts. The basic information section collected details such as gender, age, education level, years of work, and disciplinary category. The knowledge anxiety test items were rated on a Likert 5-point scale ranging from "completely disagree" to "completely agree", prompting respondents to provide a 5-level rating for each question based on their actual circumstances. Additionally, non-scoring questions and reverse-scored questions were included in the questionnaire to assess the respondents' sincerity in completing the survey, serving as a basis for subsequent data screening. During the survey distribution period, participants were tracked and reminded to ensure questionnaire recovery rates and efficiency. After two weeks, a total of 313 questionnaires were collected, yielding a response rate of 89.4%. Through manual data cleaning, 19 invalid questionnaires with completion times less than 60 s and excessively consistent answers were removed, resulting in a final dataset of 294 valid questionnaires, with an effective rate of 93.9%. This sample size met the requirements for subsequent item analysis.

Table 1 presents demographic information of the respondents. Male respondents accounted for 49.7%, while females comprised 50.3%, resulting in a gender ratio difference of only 0.7%. Regarding age distribution, the majority fell within the 36–45 age group, constituting 34%. In terms of education, the majority of surveyed research personnel possessed graduate-level education, with master's graduates comprising 49% and those with doctoral and above degrees accounting for 37.1%. Next were bachelor's degree holders and those with lower-level qualifications, constituting 13.3% and 0.7%, respectively. Concerning work experience distribution, those with 1–5 years of experience constituted 35.7%, 6–10 years accounted for 21.1%, 11–15 years constituted 16%, 16–20 years comprised 12.2%, and those with over 20 years accounted for 15%. Disciplinary categories encompassed social sciences, humanities, natural sciences, engineering, medical sciences, and others, with social sciences constituting 41.8%, humanities at 23.5%, natural sciences at 10.9%, engineering at 19%, medical sciences at 1%, and other disciplines at 3.7%. From the structure of the surveyed individuals, apart from the uneven distribution in education levels due to the influence of the research community structure, other aspects of the respondents' distribution were relatively balanced. Overall, the selection of the research sample is deemed reasonable, exhibiting certain representativeness and diversity to meet the requirements of pilot testing.

## RESULTS

### Item analysis

Item analysis, also known as item discrimination analysis, is employed to examine the discriminative ability of each measurement item and its correlation with the measured

**Table 1 Demographic information of participants.**

| Demographic Category | | Frequency | % |
|---|---|---|---|
| Gender | Male | 146 | 49.7% |
| | Female | 148 | 50.3% |
| | 18–25 Y | 65 | 22.1% |
| | 26–35 Y | 56 | 19.1% |
| Age | 36–45 Y | 101 | 33% |
| | 45–55 Y | 64 | 21.8% |
| | Under 55 years old | 8 | 2.7% |
| | College degree or below | 2 | 0.7% |
| | Bachelor's | 39 | 13.3% |
| Education | Master's | 144 | 49% |
| | Ph.D. and above | 109 | 37.1% |
| | 1–5 Y | 105 | 35.7% |
| | 6–10 Y | 62 | 21.1% |
| Work experience | 11–15 Y | 47 | 16% |
| | 16–20 Y | 36 | 12.2% |
| | 20+ Y | 44 | 15% |
| | Social Science | 123 | 41.8% |
| | Humanities | 69 | 23.5% |
| | Natural Sciences | 32 | 10.9% |
| Disciplinary background | Engineering | 56 | 19.1% |
| | Medical Sciences | 3 | 1% |
| | Other | 11 | 3.7% |

construct. The primary objective is to analyze the reliability of the scale. Item analysis is conducted during the pilot testing phase, and based on the results, modifications or deletions are made to the measurement items.

### Critical ratio method

The Critical Ratio (CR) method, also known as the Extreme Value method, is a commonly used approach for item discrimination analysis. The process involves first calculating the total scores presented by participants on the knowledge anxiety scale. Subsequently, participants are sorted in descending order based on their total scores, and the observed values of the top and bottom 27% of participants are used as the critical points for high and low groups. Participants scoring in the top 27% ($\geq 79$ points) constitute the high-score group, while those scoring in the bottom 27% ($\leq 65$ points) form the low-score group (*Wu et al., 2015*). The difference in the average scores between high and low groups is considered as the discrimination coefficient for the test item. A $t$-test is then conducted to assess whether the differences between high and low groups on each item are significant. If the critical value of the $t$-test ($t$-value) is $<3$ or the $p$-value is $>0.05$, it indicates a low discrimination for that item, and consideration should be given to its deletion. The results of the Critical Ratio method are presented in Table 2. Among the 22 test items, items Q1 and Q18 exhibit t-values (CR values) below 3 and $p$-values above 0.05, indicating their

**Table 2 Summary of project analysis results.**

| Item | Critical ratio method | Item-total correlation method | Reliability analysis method | Factor analysis method | | Number of items below standard | Disposal results |
|---|---|---|---|---|---|---|---|
| Criterion | CR(t) | Correlation coefficient | $\alpha$ | Communality | Factor loading | | |
| | <3 | <0.3 | >0.853 | <0.2 | <0.4 | | |
| Q1 | −1.9 | .142* | 0.857 | 0.007 | 0.081 | 4 | Delete |
| Q2 | −3.547 | .153** | 0.858 | 0.005 | 0.074 | 3 | Delete |
| Q3 | −8.125 | .521** | 0.846 | 0.278 | 0.527 | 0 | Reserve |
| Q4 | −12.526 | .643** | 0.841 | 0.421 | 0.649 | 0 | Reserve |
| Q5 | −8.465 | .559** | 0.845 | 0.318 | 0.564 | 0 | Reserve |
| Q6 | −10.705 | .606** | 0.842 | 0.371 | 0.609 | 0 | Reserve |
| Q7 | −13.111 | .672** | 0.839 | 0.461 | 0.679 | 0 | Reserve |
| Q8 | −11.374 | .562** | 0.844 | 0.341 | 0.584 | 0 | Reserve |
| Q9 | −11.998 | .690** | 0.839 | 0.523 | 0.723 | 0 | Reserve |
| Q10 | −10.432 | .589** | 0.843 | 0.383 | 0.619 | 0 | Reserve |
| Q11 | −13.57 | .665** | 0.84 | 0.49 | 0.7 | 0 | Reserve |
| Q12 | −12.776 | .712** | 0.838 | 0.542 | 0.736 | 0 | Reserve |
| Q13 | −16.185 | .715** | 0.838 | 0.54 | 0.735 | 0 | Reserve |
| Q14 | −12.774 | .655** | 0.84 | 0.44 | 0.663 | 0 | Reserve |
| Q15 | −4.14 | .300** | 0.854 | 0.066 | 0.256 | 1 | Reserve |
| Q16 | −6.235 | .380** | 0.851 | 0.123 | 0.351 | 1 | Reserve |
| Q17 | −3.444 | .234** | 0.855 | 0.044 | 0.21 | 3 | Delete |
| Q18 | −1.829 | 0.113 | 0.863 | 0 | 0.02 | 4 | Delete |
| Q19 | −3.44 | .284** | 0.854 | 0.079 | 0.281 | 3 | Delete |
| Q20 | −6.186 | .382** | 0.851 | 0.133 | 0.364 | 1 | Reserve |
| Q21 | −9.311 | .532** | 0.846 | 0.273 | 0.522 | 0 | Reserve |
| Q22 | −8.757 | .508** | 0.847 | 0.24 | 0.489 | 0 | Reserve |

**Notes.**

Underscores are used to indicate values that do not meet the specified criteria.

*$p < 0.05$; **$p < 0.01$.

low discrimination. Therefore, modifications or deletions should be considered for these items. The remaining items show t-values above 3, suggesting good discrimination, and thus, they are retained.

### Item-total correlation method

The item-total correlation analysis uses homogeneity testing as the basis for individual item selection. This method involves calculating the product-moment correlation coefficients between each item and the total score to determine the consistency between each item and the overall measured construct. Generally, higher correlations indicate greater homogeneity, implying a stronger relationship between the item and the construct measured by the scale, while lower correlations suggest weaker relationships. This study utilized Pearson correlation coefficients to analyze the relationships between each item and the total score. Items with a correlation coefficient >0.3 (*Yan, 2014*) and a *p*-value

<0.05 were considered satisfactory in terms of relevance to the measured construct. If the correlation coefficient was <0.3 or the *p*-value was ≥0.05, the item was deemed to have low relevance to the overall scale and was considered for deletion. Additionally, if the correlation coefficient exceeded 0.85, it indicated multicollinearity issues and did not meet the data requirements for subsequent factor analysis. The results of the Item-Total Correlation method are presented in Table 2. The analysis indicates that five items (Q1, Q2, Q17, Q18, and Q19) have correlation coefficients with the total scale score below 0.3, suggesting low relevance. Therefore, consideration should be given to deleting these measurement items. The Pearson correlation coefficients for the remaining items fall within the standard range of 0.40–0.85, with *p*-values below 0.05, meeting the specified criteria.

### Reliability analysis method

Reliability represents the consistency and stability of the scale. Reliability analysis assesses the impact of each item on the overall reliability of the scale by calculating the internal consistency. Generally, if removing a certain item improves the overall reliability of the scale, it indicates that the item has a negative impact on the scale's reliability, suggesting low internal consistency with other items. This study employed Cronbach's $\alpha$ coefficient as the reliability indicator for the scale. In general, a Cronbach's $\alpha$ coefficient >0.7 indicates good overall reliability. The reliability analysis results are presented in Table 2. The overall Cronbach's $\alpha$ coefficient for the 22 items is 0.853. After removing items Q1, Q2, Q17, Q18, and Q19 separately, the overall reliability of the scale improves. This implies that these items have weak homogeneity with the other items and should be considered for deletion.

### Factor analysis method

The factor analysis method involves examining item communality and factor loadings to determine whether to retain or discard items. Communality represents the variance that an item can explain in relation to a common latent construct, while factor loading indicates the degree of correlation between an item and a factor (*Wang, Chen & Huang, 2019*). In this study, principal component analysis was used as the factor extraction method, with the restriction of extracting only one factor, implying the existence of a single latent construct. In this context, higher communality values for items indicate a greater ability to predict the latent construct. The factor analysis method requires that the retained items should have communality values greater than 0.2 and factor loadings higher than 0.4; items failing to meet these criteria should be considered for deletion. The results of the factor analysis method are presented in Table 2.

The analysis reveals that items Q1, Q2, Q15, Q16, Q17, Q18, Q19, and Q20, a total of eight items, did not meet the specified criteria for both communality and factor loading. This indicates that these items have a lower degree of correlation with the common factor and weak homogeneity with the overall scale. Therefore, consideration should be given to their deletion.

Summary of item analysis results: Upon synthesizing the results from the four methods, items Q1, Q2, Q17, Q18, and Q19 consistently displayed multiple indicators falling below the specified standards. Consequently, these items were removed. Items Q15, Q16, and

**Table 3 KMO and Bartlett's test.**

| | | |
|---|---|---|
| **KMO (Kaiser-Meyer-Olkin) sampling adequacy measure** | | **.862** |
| Bartlett's sphericity test | Approximate Chi-Square | 1961.665 |
| | Degrees of Freedom | 136.000 |

Q20 had only one testing parameter that did not meet the specified criteria. After thorough deliberation and consensus within the research team regarding the rationale and value of these items, it was decided to retain them. In conclusion, a total of 17 items were retained for subsequent exploratory factor analysis.

## Exploratory factor analysis

Following the completion of item analysis, exploratory factor analysis (EFA) was conducted on the retained 17 measurement items to examine the structural validity of the scale. Structural validity assesses whether the dimensions of the scale align with theoretical assumptions and whether each measurement item effectively reflects the intended meaning of each dimension. EFA, a multivariate statistical method, allows the reduction of multiple observed variables into a few latent common factors, revealing the underlying structure and relationships within the data.

### KMO and Bartlett's test

Prior to conducting EFA, an adequacy test was performed on the data to determine its suitability for factor analysis. This study employed two indicators, the Kaiser-Meyer-Olkin (KMO) measure of sampling adequacy and Bartlett's sphericity test. The KMO measure reflects the proportion of variance in the data that is common among variables, with a range of 0 to 1. A KMO value above 0.7 is generally considered suitable for factor analysis (*Zhu, Ma & Feng, 2019*). Bartlett's sphericity test examines whether variables in the dataset are independent, with the null hypothesis assuming a correlation matrix as an identity matrix (no correlation among variables) (*Lei & Liu, 2013*). If the $p$-value of the test is less than 0.05, the null hypothesis is rejected, indicating the presence of common factors. KMO and Bartlett's test were conducted on the pretest data, resulting in a KMO value of 0.862, exceeding the threshold of 0.7, and a significant $p$-value for Bartlett's test. These results suggest that the items in the pretest data are not mutually independent and are suitable for exploratory factor analysis. Specific test results are presented in Table 3.

### Factor analysis

In conducting factor analysis, the present study utilized the principal component analysis (PCA) as the factor extraction method. To enhance interpretability and distinctiveness, Kaiser normalization followed by Varimax rotation, a Kaiser's normalizing approach for maximum variance, was employed to ensure orthogonality among factors (*Lei & Liu, 2013*).Without restricting the number of factors to be extracted, an initial exploratory factor analysis revealed four common factors with eigenvalues greater than 1, collectively explaining 60.72% of the total variance—consistent with the expected number of dimensions (*Zhao & Cao, 2014*). Simultaneously, an analysis of factor loadings for each

**Table 4  Explained total variance in factor analysis.**

| Component | IE | | | SSEL | | | SSRL | | |
|---|---|---|---|---|---|---|---|---|---|
| | T | PV | CP% | T | PV | CP % | T | PV | CP% |
| 1 | 5.443 | 34.021 | 34.021 | 5.443 | 34.021 | 34.021 | 3.047 | 19.042 | 19.042 |
| 2 | 1.798 | 11.238 | 45.259 | 1.798 | 11.238 | 45.259 | 2.459 | 15.366 | 3,409 |
| 3 | 1.482 | 9.265 | 5,524 | 1.482 | 9.265 | 5,524 | 2.319 | 1,494 | 48.903 |
| 4 | 1.066 | 6.665 | 61.189 | 1.066 | 6.665 | 61.189 | 1.966 | 12.286 | 61.189 |
| 5 | .880 | 5.500 | 66.689 | | | | | | |
| 6 | .725 | 533 | 71.222 | | | | | | |
| 7 | .696 | 348 | 75.570 | | | | | | |
| 8 | .626 | 3.914 | 79.484 | | | | | | |
| 9 | .578 | 3.615 | 83.099 | | | | | | |
| 10 | .507 | 3.169 | 86.268 | | | | | | |
| 11 | .499 | 3.120 | 89.389 | | | | | | |
| 12 | .447 | 2.793 | 92.181 | | | | | | |
| 13 | .381 | 2.378 | 9,560 | | | | | | |
| 14 | .356 | 2.226 | 96.786 | | | | | | |
| 15 | .341 | 2.130 | 98.915 | | | | | | |
| 16 | .174 | 1.085 | 100.000 | | | | | | |

Extraction method: principal component analysis.

measurement item across the common factors indicated that all loadings exceeded 0.4. No instances of high cross-loadings across multiple factors were observed, indicating a high level of correlation and discrimination between measurement items and common factors.

To ensure the rationality of the factor structure, the study considered various indicators, including initial eigenvalues, cumulative variance contribution, scree plot, and factor loadings, to determine the number of factors and select items for retention or deletion. Criteria for determining the number of factors included initial eigenvalues exceeding 1, cumulative variance contribution exceeding 60%, and each factor containing no fewer than 3 items. Criteria for item retention or deletion involved removing items with factor loadings below 0.4, items with absolute differences in cross-factor loadings less than 0.1, and items inconsistent with the predefined dimensions that could cause confusion in other dimensions.The component matrix of the item after adjustment is shown in Table 4.

Following these criteria, the item Q13, inconsistent with the predefined dimensions, was initially removed. Subsequent rounds of exploratory factor analysis led to the final component matrix with 16 items across 4 factors, achieving a cumulative contribution rate of 61.19%, as presented in Table 5. This indicates that the extracted common factors can explain a significant portion of the variance.

### Determine factor structure and naming

Through multiple rounds of exploratory factor analysis, the factor structure of the Knowledge Anxiety Scale for Researchers was found to be largely consistent with the four dimensions initially determined in the preliminary scale. Consequently, based on

**Table 5 Rotated component matrix.**

| NO | Items: | Components factors | | | |
|---|---|---|---|---|---|
| | | Facs 1 | Facs 2 | Facs 3 | Facs 4 |
| Q11 | I often worry that my knowledge achievements may not be recognized, and I might face questioning or criticism from others. | 0.782 | | | |
| Q10 | Sometimes I doubt the practical significance of the experiments and papers I conduct. | 0.755 | | | |
| Q8 | I often encounter professional terms or complex concepts that are difficult to understand during my studies. | 0.64 | | | |
| Q9 | I often worry that the knowledge I possess may not keep up with the latest developments in the research field. | 0.63 | | | |
| Q12 | I often worry that the literature I find may not be comprehensive, and I might miss some important knowledge. | 0.628 | | | |
| Q7 | I sometimes feel lost because the knowledge I acquire cannot be monetized. | 0.51 | | | |
| Q4 | I frequently do not know where to find knowledge relevant to my research. | | 0.705 | | |
| Q5 | Sometimes I do not know how to transform the envisioned research content into a paper. | | 0.635 | | |
| Q3 | I often feel that my knowledge structure is not comprehensive and balanced, causing anxiety about the gap between myself and others. | | 0.634 | | |
| Q6 | The rapid rate of forgetting learned knowledge sometimes makes me feel irritated. | | 0.59 | | |
| Q21 | I sometimes feel discouraged because of the high cost of knowledge consulting. | | | 0.875 | |
| Q20 | I often feel isolated on the research path. | | | 0.845 | |
| Q22 | Platforms such as WeChat and academic new media pushing knowledge-paid courses make me feel anxious and uneasy. | | | 0.645 | |
| Q16 | I can discern and filter out high-quality knowledge from the vast ocean of information. | | | | 0.826 |
| Q15 | I can flexibly use existing knowledge to propose innovative theoretical viewpoints. | | | | 0.795 |
| Q14 | I can clearly distinguish differences between knowledge in different domains. | | | | 0.655 |
| | Factor Naming | Inner emotion | Behavioral tendency | Cognitive context | Ability structure |

Extraction method: principal component analysis

Rotation method: Caesar's normalized maximum variance method.

retaining the predefined dimension names and combining the theoretical meanings of each item, the factors were named and interpreted as follows:

Factor 1: Intrinsic emotion

Comprising items Q7, Q8, Q9, Q10, Q11, Q12, Factor 1 primarily reflects researchers' intrinsic emotional experiences when facing knowledge anxiety during the processes of knowledge acquisition, internalization, and production.

Factor 2: Behavioral tendency

Encompassing items Q3, Q4, Q5, Q6, Factor 2 predominantly reflects the behavioral tendencies of researchers in the knowledge acquisition, internalization, and production processes when experiencing knowledge anxiety.

Factor 3: Cognitive context

Including items Q20, Q21, Q22, Factor 3 mainly reflects the cognitive context in which researchers find themselves, specifically, the external environment's impact on researchers' cognition, leading to the generation of knowledge anxiety.

Factor 4: Capability structure

Comprising items Q14, Q15, Q16, Factor 4 primarily reflects how researchers handle their own knowledge and competency structures in response to knowledge anxiety.

The specific factor structure and naming are presented in Table 5.

# FORMAL SCALE ADMINISTRATION

Following the completion of project analysis and exploratory factor analysis on the initial scale, this study conducted a formal implementation test of the Knowledge Anxiety Scale for Researchers to ensure its reliability and stability. The primary goals were to assess the scale's validity, reliability, and the rationality of its dimensional structure.

## Data collection

During the formal scale administration, the scope and number of participants were further expanded while maintaining the overall consistency of the test subjects. Similar to the preliminary test, data collection was carried out through online surveys. The revised Knowledge Anxiety Scale for Researchers, which underwent modifications based on project analysis and exploratory factor analysis, was uploaded to the Wenjuanxing platform to generate questionnaire links. These links were distributed nationwide through social media platforms targeting researchers.

The survey questionnaire consisted of basic information about the participants and the 16 items retained after exploratory factor analysis. Additionally, non-scoring and reverse-scoring questions were included to verify the participants' attentiveness. Over the course of one month, a total of 350 questionnaires were collected, achieving a response rate of 87.5%. After manual data cleaning, 321 valid questionnaires were obtained, resulting in a validity rate of 91.7%, meeting the requirements for the sample size in confirmatory factor analysis.

Table 6 presents demographic information about the participants, covering gender distribution, age groups ranging from 18 to 55 years, educational background, and disciplines. The sample structure remained consistent with the earlier test, aligning with

**Table 6 Demographic information of the participants.**

| | Demographic category | Frequency | % | Demographic category | | Frequency | % |
|---|---|---|---|---|---|---|---|
| Gender | Male | 164 | 51.1% | Work experience | 1–5 Y | 114 | 35.5% |
| | Female | 157 | 48.9% | | 6–10 Y | 65 | 20.2% |
| | 18–25 yrs | 69 | 21.5% | | 11–15 Y | 52 | 16.2% |
| | 26–35 yrs | 63 | 19.6% | | 16–20 Y | 43 | 13.4% |
| Age | 36–45 yrs | 110 | 33% | | 20 + Y | 47 | 16% |
| | 45–55 yrs | 71 | 22.1% | Disciplinary background | Social Science | 131 | 40.8% |
| | Under 55 years old | 8 | 2.5% | | Humanities | 74 | 23.1% |
| | College degree or below | 4 | 1.2% | | Natural Sciences | 37 | 11.5% |
| | Bachelor's | 51 | 15.9% | | Engineering | 64 | 19.9% |
| Education | Master's | 153 | 47.7% | | Medical Sciences | 4 | 1.2% |
| | Ph.D. and above | 113 | 35.2% | | Other | 11 | 3.4% |

the characteristics of the researcher population and demonstrating a degree of typicality and representativeness, meeting the requirements for the formal scale administration.

# DISCUSSION

## Reliability analysis

Following the modification of scale items during the preliminary test and the determination of the dimensions of the Knowledge Anxiety Scale for Researchers, the next step is to conduct a reliability analysis to assess the scientific rigor and applicability of the scale. Reliability analysis evaluates the stability and consistency of a scale, examining whether it produces consistent measurement results under different conditions. This analysis is a fundamental step in assessing the quality of a scale and serves as the basis for other testing procedures.

There are four commonly used methods for reliability analysis: test-retest method, parallel test method, split-half method, and internal consistency method. In this study, the internal consistency method was employed to examine the reliability of the Knowledge Anxiety Scale for Researchers. This involves calculating the correlation coefficients among the scale items. The evaluation is based on Cronbach's $\alpha$ coefficient for the overall scale and each sub-dimension, along with the Corrected Item-Total Correlation (CITI) value. A generally accepted criterion is that Cronbach's $\alpha$ coefficient should be $>0.7$, and CITI value should be $>0.4$, indicating good reliability (Zhang, 2021).

Furthermore, after renumbering the items modified during the preliminary test, and clarifying the affiliation of each measurement item to its respective dimension, the reliability of the overall scale and individual dimensions was tested using the 321 valid data collected during the formal scale administration. The results indicate an overall reliability of 0.883, with individual dimension reliabilities ranging from 0.765 to 0.862. These values meet the reliability standards, demonstrating a high level of internal consistency and reliability for the scale. Specific reliability analysis results are presented in Table 7.

**Table 7  Official scale reliability analysis results.**

| Dimension | NO | CITI | Cronbach's $\alpha$ |
|---|---|---|---|
| | M1 | 0.694 | |
| | M2 | 0.677 | |
| | M3 | 0.607 | |
| Inner emotion | M4 | 0.702 | 0.862 |
| | M5 | 0.634 | |
| | M6 | 0.615 | |
| | B1 | 0.551 | |
| Behavioral tendency | B2 | 0.683 | 0.809 |
| | B3 | 0.61 | |
| | B4 | 0.661 | |
| | C1 | 0.759 | |
| Cognitive context | C2 | 0.698 | 0.800 |
| | C3 | 0.497 | |
| | A1 | 0.653 | |
| Ability structure | A2 | 0.587 | 0.765 |
| | A3 | 0.554 | |
| Overall Reliability of the Scale | | | 0.883 |

## Model fit assessment

Model fit assessment involves validating the structural model of the scale to determine whether it aligns with the collected data. In exploratory factor analysis, the number of factors and their relationships with respective items are not explicitly defined. Therefore, the dimensions established based on this necessitate further examination of stability and reliability through confirmatory factor analysis (CFA) with real data. CFA, conducted under the assumptions of a predetermined number of factors and their relationships with items, evaluates the fit between the factor model created during exploratory factor analysis and the actual data.

In this study, structural equation modeling (SEM) was employed as the method for model fit assessment. Utilizing data from the formal administration of the scale and AMOS 26.0 software, the SEM assessed the model fit. SEM is a multivariate statistical method that combines factor analysis and path analysis, considering relationships between observed variables and latent variables, as well as relationships among latent variables.

In confirmatory factor analysis, the measurement items of the scale are treated as observed variables, and the factor dimensions are treated as latent variables. A path model was constructed with four latent variables and sixteen observed variables. Various fit indices were used to evaluate the degree of fit between the actual data and the factor model. Poor fit between data and the model suggests the need for further adjustments to the dimensions and items of the scale.

For model fit assessment, three types of indices were primarily utilized: absolute fit indices, incremental fit indices, and parsimony fit indices. These indices provide a comprehensive evaluation of the model's fit. Absolute fit indices, such as GFI and RMSEA,

**Table 8  Results of model fit tests.**

| Model | $x^2$/df | RMSEA | GFI | CFI | IFI | PCFI | PNFI |
|---|---|---|---|---|---|---|---|
| Adaptation Standards, | <3 | <008 | >0.9 | >0.9 | >0.9 | >0.5 | >0.5 |
| Results | 1.983 | 0.055 | 0.930 | 0.956 | 0.956 | 0.780 | 0.747 |

reflect the magnitude of residuals between the model and data. Incremental fit indices, including CFI and IFI, indicate the improvement of the model compared to a baseline or nested model. Parsimony fit indices, like $x^2$/df, PCFI, and PNFI, consider the model's complexity and the number of free parameters.

Generally, good model fit is indicated by $x^2$/df < 3, RMSEA < 0.08, and GFI, CFI, IFI > 0.9, PCFI, PNFI > 0.5 . According to the results of the model fit assessment in Table 8, $x^2$/df is 1.983, RMSEA is 0.055, GFI is 0.93, CFI is 0.956, IFI is 0.956, PCFI is 0.780, and PNFI is 0.747. All these values meet the aforementioned fit criteria, indicating a good fit between the model and the data. The standardized path diagram for confirmatory factor analysis is presented in Fig. 1.

## Validity analysis

Validity refers to the extent to which a scale can effectively measure the intended concept or trait, indicating whether the items of the scale genuinely reflect the characteristics or states of the measured subjects. Validity analysis is a crucial indicator for evaluating the usefulness of a scale, forming the foundation and assurance for the scale's promotion and application. In this study, validity testing for the Knowledge Anxiety Measurement Scale for Researchers was conducted from three aspects: content validity, convergent validity, and discriminant validity.

Content validity: Content validity assesses whether the measurement content of the scale sufficiently covers the range of the measured concept. This study adhered strictly to the grounded theory procedures in forming the structural dimensions of knowledge anxiety for researchers. Based on this, an initial measurement item pool was developed by combining existing tools for measuring research data anxiety and information anxiety. The research team thoroughly examined the semantics, language, and logic of the measurement items. Furthermore, five experts and scholars with relevant research backgrounds were invited for additional item review and suggestions. Adjustments and modifications were made to items showing insufficient representation and cross-dimensionality. Consequently, the developed measurement scale in this study underwent a standardized development process, ensuring that it adequately covers the conceptual scope of knowledge anxiety for researchers and exhibits good content validity.

Convergent validity: Convergent validity assesses the correlation between measurement indicators of the measured concept. Higher correlations between measurement indicators result in higher factor loadings, indicating a better convergence effect of the measured concept. The study primarily assessed the convergent validity of the scale by calculating standardized factor loadings, average variance extracted (AVE), and composite reliability for each factor. The data revealed that the standardized factor loadings for each factor ranged

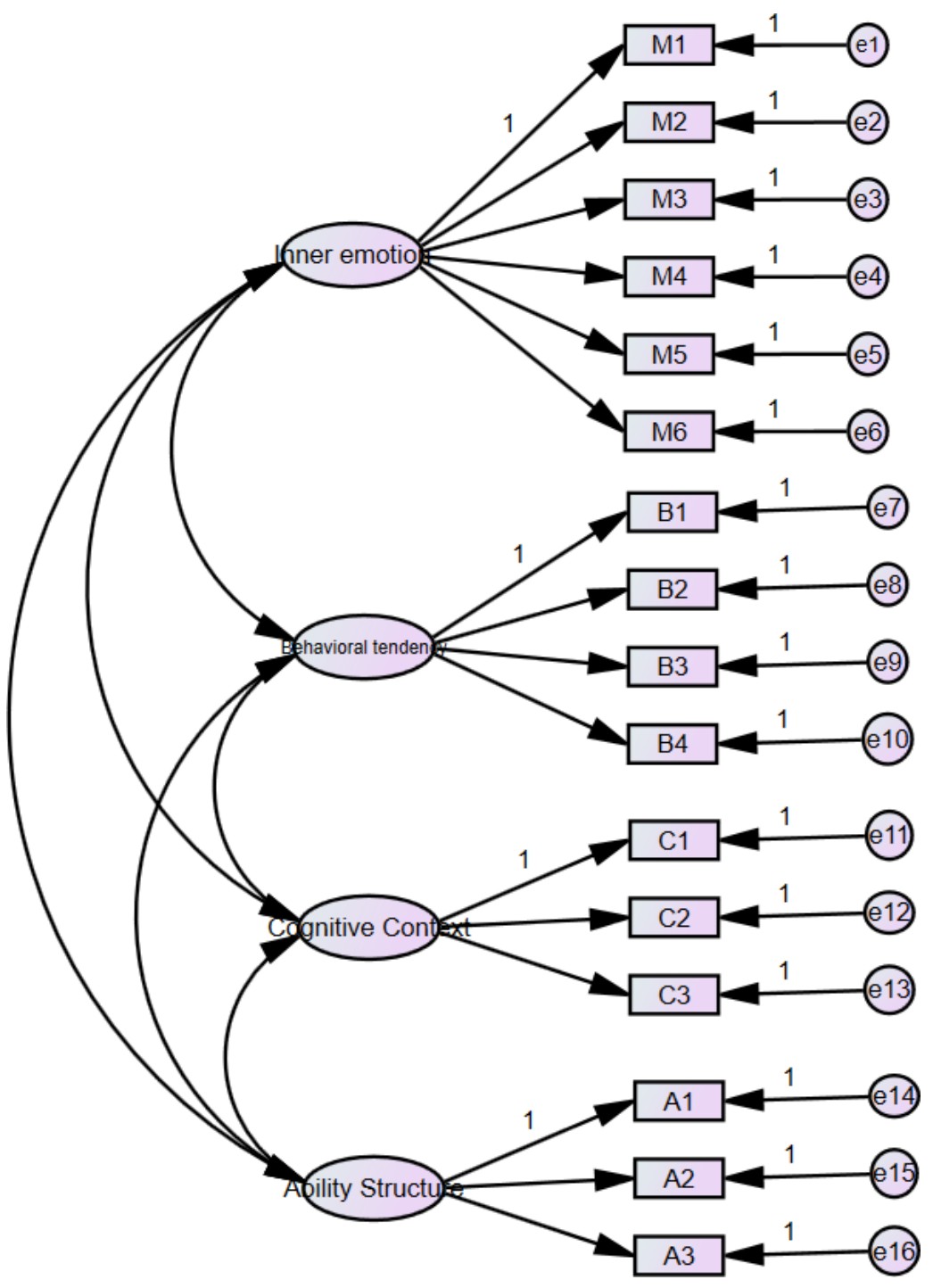

**Figure 1   Standardized path diagram for confirmatory factor analysis.**

from 0.549 to 0.887, all exceeding the 0.50 standard, indicating that the measurement items effectively reflect the same latent construct. The AVE for each factor ranged from 0.514 to 0.614, all exceeding the 0.5 threshold, demonstrating a high degree of explanatory power

**Table 9  Presents the analysis results of convergent validity.**

| Items | | Standardized factor loadings | AVE | CR | CR |
|---|---|---|---|---|---|
| Inner emotion | M1 | 0.754 | | | |
| | M2 | 0.728 | | | |
| | M3 | 0.669 | 0.514 | 0.864 | |
| | M4 | 0.763 | | | |
| | M5 | 0.703 | | | |
| | M6 | 0.679 | | | |
| Behavioral tendency | B1 | 0.648 | | | |
| | B2 | 0.777 | 0.522 | 0.813 | |
| | B3 | 0.709 | | | |
| | B4 | 0.750 | | | |
| Cognitive context | C1 | 0.887 | | | |
| | C2 | 0.867 | 0.614 | 0.821 | |
| | C3 | 0.549 | | | |
| Ability structure | A1 | 0.812 | | | |
| | A2 | 0.710 | 0.53 | 0.770 | |
| | A3 | 0.653 | | | |

of each factor for the variance in its measurement items. The composite reliability for each factor ranged from 0.77 to 0.864, all meeting the recommended standard of 0.6, indicating a high level of correlation among the measurement items and high consistency in the measured latent construct. Considering these indicators collectively, the developed scale in this study confirmed good convergent validity. The results of the convergent validity analysis are presented in Table 9.

Discriminant validity assesses whether different latent factors can be distinguished from each other, meaning that a measurement item should only reflect one latent factor, and there should be no measurement items crossing different factors. Different latent factors should exhibit sufficient distinctiveness. Discriminant validity is primarily assessed by determining whether the square root of the average variance extracted (AVE) for each latent factor is greater than the squared correlation between that factor and other latent factors.

The results indicate that the square roots of the average variance extracted values for the four latent factors are 0.717, 0.723, 0.784, and 0.728, respectively. All these values exceed the squared correlation values between the corresponding factor and other latent factors. This suggests that the Knowledge Anxiety Measurement Scale for Researchers developed in this study possesses satisfactory discriminant validity. The results of the discriminant validity analysis are presented in Table 10.

Drawing inspiration from relevant expressions on knowledge anxiety found in existing scales, an initial pool of items is constructed. The research team reviews and assesses these items with expert consultations to form the initial measurement scale. Subsequently, a pre-test is conducted, and item analysis along with exploratory factor analysis is employed to

**Table 10 Test results for discriminant validity.**

|  | Inner emotion | Behavioral tendency | Cognitive context | Ability structure |
|---|---|---|---|---|
| Inner emotion | (0.717) |  |  |  |
| Behavioral tendency | 0.646 | (0.723) |  |  |
| Cognitive context | 0.476 | .481 | (0.784) |  |
| Ability structure | 0.106 | 0.311 | 0.091 | (0.728) |

refine and streamline the scale items. Finally, the confirmatory factor analysis is performed using AMS 26.0 software to validate the factor structure model. The validity of the scale is examined from three aspects: content validity, convergent validity, and discriminant validity. The end result is a formally structured scale consisting of four dimensions and 16 items, which is rigorous and scientifically sound.

## CONCLUSIONS

This investigation, despite its innovative approach to examining knowledge anxiety among environmental scientists, encounters certain constraints. The primary limitation lies in the geographical and cultural specificity of our study's sample, which may affect the universality of our findings. The reliance on self-reported data introduces a potential for bias, such as social desirability or recall inaccuracies, which could skew the results. Furthermore, the cross-sectional design of our research inhibits our ability to establish causality between knowledge anxiety and its influence on productivity and mental health in the environmental science domain .

Future studies should endeavor to validate the knowledge anxiety scale in a broader range of cultural and geographical contexts to ensure wider applicability. Employing longitudinal research designs would aid in unraveling the causal relationships between knowledge anxiety and its impact, offering a dynamic perspective on this psychological phenomenon within the environmental science community . Incorporating objective measures alongside self-reported data could mitigate bias, thereby enhancing the reliability of findings related to research productivity and mental health . Furthermore, examining knowledge anxiety within interdisciplinary research teams could illuminate the variegated experiences across scientific disciplines, potentially uncovering discipline-specific challenges and coping mechanisms. Lastly, the development and assessment of targeted interventions to alleviate knowledge anxiety among environmental scientists stand as an imperative direction for future research. Such interventions could significantly contribute to improving both the well-being of researchers and the efficacy of environmental science in addressing the planet's most pressing issues.

## ACKNOWLEDGEMENTS

The support and resources provided by the Qilu University of Technology Library have been invaluable, and for that, we are immensely grateful.

### Funding

The authors received no funding for this work.

### Competing Interests

The authors declare there are no competing interests.

### Author Contributions

- Yu Zhenlei conceived and designed the experiments, performed the experiments, prepared figures and/or tables, and approved the final draft.
- Lin Song analyzed the data, authored or reviewed drafts of the article, and approved the final draft.
- Dong Minyi analyzed the data, authored or reviewed drafts of the article, and approved the final draft.
- He Qiang conceived and designed the experiments, performed the experiments, prepared figures and/or tables, and approved the final draft.

### Human Ethics

The following information was supplied relating to ethical approvals (*i.e.*, approving body and any reference numbers):

Department of Science and Technology, Linyi University.

### Data Availability

The raw measurements are available in the Supplementary Files.

### Supplemental Information

Supplemental information for this article can be found online at http://dx.doi.org/10.7717/peerj.18478#supplemental-information.

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
