# Peer review of "Assessing knowledge anxiety in researchers: a comprehensive measurement scale"

_PeerJ, doi:10.7717/peerj.18478_

## Round 0.1 · original submission · Minor Revisions

Thank you for your interesting work. We have comments from two reviewers which suggest ways of improving your work. I think that their comments are clear and that each justifies a response, either indicating how the manuscript has been changed in response or providing an explanation as to why you feel no changes are needed for that point. I look forward to seeing a revised version of your manuscript, with changes clearly indicated, along with a response to/rebuttal of each point from our reviewers.

Reviewer 1 ·

Basic reporting

Appropriate

Experimental design

Appropriate

Validity of the findings

Appropriate

Additional comments

Dear Authors,

Information anxiety among researchers continues to be a significant concern in our increasingly information-rich world. I commend you for addressing this crucial issue. Your study is a valuable contribution to the literature and has been meticulously prepared.

However, I believe there are several important issues that need to be addressed before publication. My suggestions are listed below:

The introduction is too long and disorganized. Please shorten and structure it more clearly.
There is a considerable amount of repetition throughout the text. Simplify and streamline these sections.
Ensure adherence to spelling rules throughout the text.
In the discussion section, the method is explained. These sections should be moved to the methods section.
Under the tables, specify the abbreviations used.
Pay attention to capitalization in table titles (e.g., Table 9).
Avoid splitting words in tables (e.g., Table 6).

good work

·

Basic reporting

The article was presented professionally in terms of grammar and structure and to a large extent clarity. However, there was insufficient background provided at the introductory level. The introduction should introduce the key concept under discussion, which appears to be a deficit. Lines 47-52 in the introduction are best expunged or best placed in the conclusion as it is already present a conclusion where an introduction should have been. Very commendable is the fact that ideas presented in the development of the initial item in 2.1 give a clear direction as to the theoretical and conceptual underpinnings of the key variable under study, however, they were not reported in a style that makes it easy to understand. It is best if these key variables are explained in more detail. Some of the ideas presented in the beginning part of the Development of the initial items under materials and methods would have been more aped if they were introduced in the introductory section. Also, the introduction contained some repetitive discussions. Lines 62 to 119 contain redundant and repetitive discussions. This should be streamlined to be more concise.

Furthermore, the article can greatly benefit from the introduction of objectives research questions, or hypotheses. The headings of the discussion sections can serve as a guide for this. The article also contained sufficient literature and good references generally. This can however be strengthened. It would be best if previous studies quoted in line 42 contained more than one study to justify that claim. Also, proper citations need to be done to back up the claims especially lines 124 to line129. The author cited in line 130 needs to be properly cited. Similarly, citations need to be properly done for line 179.

The raw data shared provides research transparency that provides more clarity to the result obtained

Experimental design

The work contained a rigorous investigation performed to a high standard, with the methods described with sufficient detail & information to replicate. The three-dimensional structural model of researchers’ knowledge anxiety gives a strong conceptual backing to the instrument constructed. Related literature that abounds on these areas can be cited to strengthen this further. The introduction of subheadings in materials and methods in 2.1 can help with the flow of information and categorization of related themes for easy comprehension. This can be done by breaking 2.1 into stages or steps to clearly show what was done at each stage of the development of the initial items. The same can be done for 2.2.

The report on the pilot study and results reveals a detailed and robust analysis required for test development and validation. It is commendable that the authors were able to give brief explanations for what each psychometric property means before giving details and results obtained. For the dimensions of the instrument, there needs to be consistency or reconciliation or clearer details of the knowledge anxiety structural dimensions outlined in section 2.1 of cognitive, emotional, and behavioral with the 4 dimensions mentioned in 2.2 in item consolidation and refinement and the 4-factor structure and naming in 3.2.3. For more clarity, more details need to be supplied as to how the four factors structure of Knowledge anxiety obtained from multiple rounds of factor analysis and item consolidation falls or relates with the three dimensions outlined in 2.2. For instance, where do items under capability structure fall within the conceptualized three structural dimension.

Validity of the findings

The statistical detail for the article is generally sound and high. The statistical detail on the items that were deleted at each section and the rationale is commendable However, for greater clarity, it would be great if there is a tabular presentation that shows this. For instance, a tabular presentation that would carry each factor structure, the initial items under each factor structure, and the items that were deleted at each structure/factor.

The article contained a robust and statistically sound discussion on reliability, model fit, and validity analysis. However, these discussions can further be strengthened if they are substantiated with citations. For instance, what was done in line 440 was great and is a great example. The article can greatly benefit from a section or paragraph that contains scoring and interpretation of the knowledge anxiety instrument. The article also has a well-stated conclusion

Additional comments

The number of participants in the main administration of the research instrument developed after the pilot testing should be mentioned in the abstract. Omitting this detail gives the impression that the analysis done was based on details from trial testing alone. The article generally presents a commendable job of developing and validating the Knowledge anxiety scale with robust psychometric properties

---

## Round 0.2 · accepted · Accept

Thank you for your revisions and rebuttal. We have no additional questions from our reviewers and I am delighted to accept your manuscript for publication. Well done!

·

Basic reporting

Great improvement on the reporting

Experimental design

Significant improvement on the design

Validity of the findings

Great improvement on the validity of findings

Additional comments

The author(s) did a commendable work based on the review feedback which has significantly improved the quality of the work. Great work